# Oligomeric Enteral Nutrition in Undernutrition, due to Oncology Treatment-Related Diarrhea. Systematic Review and Proposal of An Algorithm of Action

**DOI:** 10.3390/nu11081888

**Published:** 2019-08-13

**Authors:** Alejandro Sanz-Paris, María Martinez-García, Javier Martinez-Trufero, Julio Lambea-Sorrosal, Fernando Calvo-Gracia, María Elena López-Alaminos

**Affiliations:** 1Department of Endocrinology and Nutrition, Miguel Servet Hospital, Zaragoza 50009, Spain; 2Instituto de Investigación Sanitaria Aragón (IIS Aragon), Zaragoza 50009, Spain; 3Department of Oncology, Miguel Servet Hospital, Zaragoza 50009, Spain; 4Department of Oncology, University Clinic Hospital, Zaragoza 50009, Spain; 5Department of Endocrinology and Nutrition, University Clinic Hospital, Zaragoza 50009, Spain

**Keywords:** oncology treatment related diarrhea, review, nutritional algorithm, oligomeric enteral nutrition formula

## Abstract

Oncology treatment-related diarrhea and malnutrition appear together in oncological patients because of the disease itself, or the treatments that are administered for it. Therefore it is essential to carry out a nutritional treatment. Enteral nutrition formulas, containing peptides and medium chain triglycerides, can facilitate absorption in cases of malabsorption. There are few references to the use of enteral nutrition in the clinical society guidelines of patient management with oncology treatment-related diarrhea (OTRD). A bibliographic review of the studies with oligomeric enteral nutrition in OTRD found only nine studies with chemotherapy (all with the same oligomeric formula in which oral mucositis improves, while the rest of the outcomes show different results), and eight studies with radiotherapy (with different products and very heterogeneous results). We hereby present our action algorithm to supplement the diet of OTRD patients with an oligomeric enteral nutrition formula. The first step is the nutritional assessment, followed by the assessment of the functional capacity of the patient’s intestine. With these two aspects evaluated, the therapeutic possibilities available vary in degrees of complexity: These will range from the usual dietary recommendations, to supplementation with oral oligomeric enteral nutrition, along with complete enteral nutrition with oligomeric formula, and up to potentially total parenteral nutrition.

## 1. Introduction

In the recent years there have been substantial advances in molecularly-targeted therapies for the treatment of patients with cancer. However, chemotherapy and radiotherapy are continuously used alone, combined, or utilized even before surgery. Chemo-radiotherapy causes and even exacerbates a symptom that worsens the nutritional status of our patients, which symptom is caused by digestive toxicity, such as nausea, vomiting, mucositis and diarrhea.

Diarrhea and malnutrition appear together in the oncological patient as a consequence of the disease itself, or because of the administered treatments. Chemotherapy and radiotherapy cause different undesirable effects on the gastrointestinal tract mucosa, like inflammation, edema, ulceration and atrophy. The increase in mucosal permeability, combined with immunosuppression, predisposes the patient to bacterial translocations, septicemia and ischemia [1].

The most serious complications by chemotherapy include myelosuppression, hepatic or renal disorder, oral mucositis or diarrhea. However, prevention methods have not yet been well established. On the other hand, treatment with radiotherapy causes radiation-induced enteritis (inflammation of the small intestine)—when applied directly to the abdominal and pelvic regions in the treatment of urological, gynecological or gastrointestinal tumors. The small intestine is the most sensitive organ to radiation. Studies show that 50–75% of patients under study develop mild to severe enteritis after pelvic or abdominal radiation therapy [2]. Chronic radiation enteritis has been reported in up to 20% of patients receiving pelvic radiotherapy; with intestinal failure developing in approximately 5% [3]. Radiotherapy of the head and neck or pelvic region is associated with gastrointestinal symptoms and weight loss in up to 80% of patients.

Oncology treatment-related diarrhea (OTRD) is one the most important manifestations of mucositis induced by chemotherapy. The exact objective incidence of OTRD may not be well known, because its incidence is described according to patient complaints, and is often assessed by non-medical staff; thus, its incidence may often be underestimated. Several studies estimate that approximately 10% of patients with advanced cancer will present it, with colorectal cancer being the most frequent. Other authors report that between 20% and 40% of all patients with chemotherapy suffer severe diarrhea [4,5]. In randomized phase III trials, these agents are associated with diarrhea in 50–80% of patients, when used alone or in combination [6].

In recent years, other agents have been incorporated as oncology treatments that have direct consequences on a patient’s digestive function, for example: Tirosin-multikinase inhibitors Ssorafenib, Lenvatinib, Iimatinib, Sunitinib, Pazopanib and Cabozantinib). Many of them display new indications in several cancer subtypes, such as clear cell carcinoma, hepatocarcinoma, gastrointestinal stromal tumors, thyroid cancer and lung cancer. In most cases, diarrhea is the main and most common adverse event. Since those treatments are often administered chronically, long-term control and care of this problem and its consequences on nutritional status is mandatory [7]. Besides, in the last few years, immunotherapy has emerged as common new treatments in many other tumors, and once again diarrhea is a common adverse event whose control and treatment is going to be crucial in order be able to maintain them in long-term use [8].

Among the most frequent OTRD complications are: Dehydration, electrolyte loss, inadequate absorption of fats, lactose, bile salts and vitamins, social and personal problems, a reduction of treatment dose or discontinuation, an alteration of the intestinal flora, hemodynamic instability in critical patients and malnutrition. This represents a double declining slope: Clinical and economic, both contributing to reducing survival rates [9].

Many of these side effects of OTRD are associated with malnutrition. Nutritional disorders often occur in cancer patients because of treatments, or by being related to cancer itself, and thus exacerbate the complications and adverse effects from cancer treatments. OTRD typically leads to discomfort and/or pain, which in turn may lead to less food intake and greater susceptibility to infections. Diarrhea, anorexia and nutritional disorder are three highly correlated factors, and form a vicious circle as an adverse effect caused by chemotherapy [10].

The published prevalence of malnutrition could vary due to the different identification of malnutrition definitions, but also because of the tumor’s location and the therapy’s intensity. We can find a prevalence of malnutrition before the start of cancer therapy between 3–52%. During radiotherapy and chemotherapy, this percentage of malnourished patients rises to 44–88% [9]. The prevalence of diarrhea with new agents is variable; for example, it was measured around 53% with anti-EGFR TKI agents [11], 27–54% with anti-CTLA-4 therapy [12] and 33–49% with the tyrosine kinase inhibitor Imatinib [13].

It is recognized that malnutrition is associated with lower physical functioning, lower immune status, more severe late radiotherapy-induced toxicities, treatment interruptions and delayed, lower chemotherapy response rates, hospital readmissions, impaired quality of life, and increased mortality [14]. Therefore, it is important to prevent, recognize and treat malnutrition in an early phase of oncology treatment.

Regardless of the cause of the OTRD, it is essential to carry out a nutritional treatment that facilitates absorption, reduces waste and digestive symptoms and contributes to the prevention of malnutrition, which is so frequent in the oncological population. The approach of malnutrition prevention in this situation is critical to enable the patient to withstand prolonged oncological treatments. In patients with clinical stability and a functioning bowel, enteral nutrition can be a valid alternative to parenteral nutrition. Enteral nutrition exerts a trophic effect upon the intestinal mucosa, and improves the hepatotrophic and splanchnic flow [15]. In this way, oral enteral nutrition during oncotherapy enhances dietary intake, prevents weight loss and promotes adherence to radiotherapy [16]. In patients with cancer, parenteral nutrition has not been shown to prolong the survival of patients, and even increases the prevalence of complications compared to enteral nutrition [17].

There are different types of enteral nutrition formulas, which can be classified by their composition. The European Society for Clinical Nutrition and Metabolism (ESPEN) guidelines on the definitions and terminology of clinical nutrition affirm “formulas containing peptides and medium chain triglycerides can facilitate absorption in case of, e.g., malabsorption or short bowel syndrome” [18].

Despite its obvious usefulness, there are few references to the use of enteral nutrition in the algorithms of patient management with OTRD. This article´s objective of introducing our algorithm is organized into three parts: Firstly, we review of the main clinical practice guidelines for the treatment of OTRD nutritional recommendations to assess the role that is attributed to enteral nutrition in their algorithms. Secondly, we carried out a bibliographic review on the studies with oligomeric enteral nutrition, in patients with OTRD. Finally, we present our action algorithm to supplement the diet of patients with OTRD with an oligomeric enteral nutrition formula.

## 2. Clinical Practice Guidelines for the Management of OTRD

OTRD management standards guidelines are widely published and state that their main aim should be to reduce the volume of diarrhea, treat dehydration aggressively, and to use drugs if symptoms persist [19,20,21,22,23,24,25,26,27,28,29,30,31,32,33,34,35,36,37].

In general, all guidelines recommend that an early intervention should be performed when the patient initiates oncological treatment to prevent the progression of the severity of diarrhea. The treatment to be performed will depend upon the degree of the severity of the diarrhea, as well as the presence of other risk factors: Fever, vomiting, neutropenia, frank bleeding in the stool, moderate/severe abdominal pain, and dehydration. Patients with mild diarrhea and without risk factors can be treated on an outpatient basis with oral antidiarrheal and pharmacological measures, while those with severe diarrhea and/or risk factors will need in-patient treatment.

Although an adequate intake of fluids and early nutritional recommendations can reduce the incidence and severity of diarrhea, there are few specific recommendations in the guidelines we have found. They focus primarily on pharmacological treatment and on ruling out secondary processes that may be causing infectious diarrhea, medication, obstruction, or concurrent diseases, such as diabetes mellitus, hyperthyroidism, pancreatic insufficiency, inflammatory bowel disease.

Table 1 summarizes the dietary recommendations of the main international scientific societies [19,20,21,22,23,24,25,26,27,28,29,30,31,32,33,34,35,36,37]. All guidelines recommend an increase liquid consumption to avoid dehydration (3–4 L, with a high sodium content, such as sports drinks, broths, soups, etc.). Some societies also recommend the intake of foods rich in potassium, such as fruit juices and nectars, sports drinks, potatoes with skin and plantains [29,38,39]. In addition, in order of the higher to lower frequency of appearance in the guidelines, we found: Avoid the intake of alcohol and products with caffeine [20,21,22,23,24,25,26,29,30,31,33,34,37,38,40,41], spicy and fried foods [19,23,25,26,30,33,34,38,41], a low fat diet [19,21,22,23,26,31,32,38], and small but frequent meals [20,23,24,26,28,38,40].

Other recommendations appear infrequently because they are logical, such as avoiding very hot or cold foods and fluids [23,30,33,38], or products that contain sorbitol [23,34,38,40].

Regarding the fiber content of the diet, eight out of the twenty-three reviewed guides recommend a poor fiber diet [19,20,22,23,25,31,34,38], some specify that insoluble fiber should be avoided [19,23,25,31,38], and others recommend increasing the intake of soluble fiber [22,23,34,37,38].

The intake limitation of products containing lactose is one of the most frequently found recommendations (fourteen out of the twenty-three reviewed) [19,20,21,22,23,24,25,28,29,31,34,37,38,40], despite the recent European Society for Medical Oncology (ESMO) Clinical Practice Guidelines, which indicate that there is not enough evidence to avoid lactose in the diet of these patients [41].

It is striking that most of the treatments that appear in the guidelines are focused upon diarrhea, but very few consider the adequate nutritional support to avoid malnutrition. Diarrhea is associated with loss of water and electrolytes, but also with nutrient malabsorption. Oncological patients are susceptible to a high risk of malnutrition due to cancer anorexia-cachexia syndrome, the oncological treatment toxicity, cancer-associated digestive dysfunction, psychological factors, etc. The appear-ance of diarrhea in these types of patient substantially worsens their quality of life [10,14].

Nutritional counseling and oral nutritional supplements should be used to increase dietary intake and to prevent oncology-therapy-associated weight loss and interruption of cancer therapy [42]. ESPEN expert group recommendations for action against cancer-related malnutrition press the point that patients with cancer are at particularly high risk for malnutrition, both because of the disease and its treatments. Furthermore, the deaths of 10–20% of patients with cancer can be attributed to malnutrition, rather than to the malignancy itself. Nutrition counselling by a health care professional is regarded as the first line of nutrition therapy [15]. Ravasco et al. [43] observe that nutritional therapy provided an early and timely individualized nutritional counseling, and that education has a sustained effect upon outcomes, nutritional intake and status, late radiotherapy toxicity, quality of life and prognosis.

Certain guidelines recommend the use of some type of enteral nutrition treatment [19,22,23,27,33,36,39,41]. The majority refers to the contribution of glutamine [19,36,39,41], probiotics [30,39] and omega 3 fatty acids [36,41], although the recent ESMO Clinical Practice Guidelines [41] indicate that more studies are required to advise its use. With regard to the contribution of nutritional supplements, some only indicate that products with high osmolality should be avoided [20,34], others that these supplements should only be used in severe degrees of diarrhea [23] or in severe malnutrition [33], but only two guides refer to the use of oligomeric formulas [22,27].

It has been suggested that an oligomeric diet might be protective to the intestinal mucosa when administered to patients with diarrhea associated with oncotherapy. Polymeric enteral nutrition is the usual type of formula used in patients who need nutritional enteral support, but some patients could need alternative formulas because of severe diarrhea. Choice of oligomeric enteral formula may be important to improve nutrients’ absorption and to achieve the goal of the successful completion of scheduled nutrition and a smooth transition to the normal diet [44].

The polymeric enteral formula contains nitrogen in the form of whole proteins and needs the gastrointestinal tract digestion process. On the other hand, the nitrogen sources of oligomeric enteral formula are hydrolyzed proteins (oligopeptides of varying lengths, dipeptides and tripeptides).

These peptides of oligomeric enteral formulas have specific uptake transport mechanisms and are thought to be absorbed more efficiently than whole proteins. Given impaired digestive function in cancer patients with diarrhea associated with systemic treatment, oligomeric enteral formulas place less burden on the digestive system and are absorbed better, and thus are thought to be more favorable options compared with polymeric formulas [18].

Data from convincing animal experiments and human preclinical studies report reduced pancreatic or biliary stimulation with oligomeric formula diet, compared to a regular or polymeric diet [45]. Protein is provided as free amino acids or peptides, which reduces the requirement for enzymatic hydrolysis before absorption, and reduces the risk of presenting an ‘antigenic’ load. The provision of fats as medium chain triacylglycerol allows their absorption directly into portal blood without the need for enzymatic hydrolysis and emulsification by bile salts. Thus, oligomeric formulae can be valuable therapies in Crohn’s disease [46], and help maintain gut integrity by delivering a readily-available source of nutrients directly to the gastrointestinal mucosa. Moreover, oligomeric enteral nutrition was shown to reduce the mucosal production of pro-inflammatory cytokines such as tumor necrosis factor (TNF)-alpha and interleukin (IL)-6 in patients with Crohn’s disease, and dietary histidine ameliorates murine colitis by an inhibition of pro-inflammatory cytokine production in macrophages [47].

Mucosal injury is a common oncological treatments side effect, resulting in symptoms of stomatitis and diarrhea. The mucosal injury causes impaired absorption and eating disorder in addition to pain associated with the bleeding and ulceration, resulting in increased healthcare costs and an impaired quality of life [48].

Oligomeric enteral nutrition formulas could help to maintain mucosal integrity in the gastrointestinal tract, thereby resulting in maintained nutrient absorption. The pathogenic mechanism underlying gastrointestinal mucosal injury caused by anticancer therapies involves damage in the DNAs of normal mucosal epithelial cells and fibroblast cells, resulting in suppressed cell growth and tissue repair. Kawashima et al. [49] investigated the effects of oral supplementation with oligomeric enteral nutrition diet on mucin in 5-fluorouracil-induced intestinal mucositis. The results of their studies show that the length of the intestinal tract, dry weight and villus height were reduced by 5-fluorouracil administration. Those animals receiving an oligomeric enteral nutrition diet showed an improvement in digestive and absorptive function. The authors conclude that oligomeric enteral nutrition has the ability to improve the intestinal tract defense function mainly exerted through small intestinal mucin, indicating the possibility of relieving gastrointestinal mucosal injury caused as an adverse effect of anticancer drugs.

Supported by these studies, we infer that the use of nutritional supplements with oligomeric formulas in patients with OTRD who present malnutrition could be very useful. Prior to the design of a performance algorithm, we consider it would be necessary to carry out a literature review on the use of oligomeric enteral nutrition formulas in patients with OTRD.

## 3. Review on the Use of Oligomeric Enteral Nutrition Formulas in Malnourish Patients with OTRD

The mechanisms involved in the chemotherapy-associated diarrhea are not well explained, and they vary according to the type of chemotherapy used, being more common after the administration of topoisomerase inhibitors (irinotecan, the highest risk), methotrexate at high doses, fluoropyrimadines (5-fluorouracil, capecitabine) or taxanes as docetaxel. It has been postulated that it could be due to the damage of the intestinal crypts, or changes in the intestinal microflora, with an alteration of the transport of fluids in the colon and a lower absorption of water. In addition, mucositis induced by chemotherapy produces a malabsorption of nutrients that contribute to osmotic diarrhea. Most of the literature is based upon clinical observations in which the number and consistency of bowel movements are collected, as well as whether or not they are accompanied by blood, mucus and pain. On the other hand, irradiation on the cells of the intestinal mucosa causes a loss of mucosal surface, with the loss of intestinal permeability, motility and absorption of nutrients. It also facilitates the translocation of the intestinal microflora, the appearance of ulcers, necrosis, mucous membrane bleeding and even more serious complications, such as intestinal fistula.

The most characteristic clinical manifestation is diarrhea [50]. Lees clear are the mechanisms of TKI-induced and immunotherapy-induced diarrhea, since it has not been widely studied yet, but most of the hypotheses link this toxicity to mechanisms that are shared with chemotherapy-induced diarrhea [7].

An oligomeric enteral nutrition diet can be a nutritional treatment of choice in the patient with OTRD for its easy absorption, inhibition of pro-inflammatory cytokine production and maintenance of mucosal integrity. There are very few published reviews on the efficacy of enteral peptide nutrition in patients with diarrhea associated with oncotherapy. We found a Cochrane review in patients with radiotherapy, but the researcher was not able to perform meta-analysis due to the lack of studies [51].

Given the limited published evidence, we consider that a bibliographic review on the subject was necessary prior to the preparation of the nutritional support algorithm in patients with OTRD.

For this review, we researched on-line databases PUBMED, MEDLINE, EMBASE and the Cochrane library up to March 2019, using the following terms: ((Chemotherapy) or (radiotherapy)) and ((oral mucositis) or (diarrhea) or (peptide diet) or (elemental diet) or (oligomeric diet)) in order to collect potentially relevant articles. Animal, not clinical, trials and non-adult studies were excluded. Retrieved articles were reviewed for relevance, duplicates were discarded, and the full text of all potentially relevant articles was classified and assessed for inclusion using predetermined criteria. Only clinical trials about oligomeric elemental or peptide enteral nutrition were screened. Reference lists of all of the included articles were reviewed for additional possibly relevant citations. Further details about the literature search process are provided in Figure 1. Relevant studies were identified [52,53,54,55,56,57,58,59,60,61,62,63,64,65,66,67,68], and their results are summarized in Table 2 and Table 3.

Regarding the studies with oligomeric diets in oncological patients under treatment with radiotherapy [52,53,54,55,56,57,58,59] (Table 2), 998 patients were collected (476 in the intervention groups and 522 in the control groups). The three oldest studies [52,53,54] were conducted with Vivonex^®^ without significant differences in the response to stool characteristics, or in the amount of weight lost. The next two studies to appear [55,56] were performed with Vital HNR^®^, detecting an improvement in the severity and duration of diarrhea in the group that took the enteral nutrition product. Subsequently, another study with 677 patients [57] shows a significant reduction in intestinal toxicity, the suppression of radiotherapy and weight improvement with an undefined elemental diet. But this is a study that is only a conference abstract, and does not define the type of oligomeric enteral nutrition used. The last study that we found which compares oligomeric diet with habitual diet is that of McGough et al. [58] that uses EO28^®^. It does not find improvement in gastrointestinal symptoms or nutritional status, because only 24% of patients take the product. Finally Feng Shao et al. [59] compares an oligomeric diet (Peptisorb^®^) with or without microorganisms and fish oil, with an improvement in abdominal pain, bloating and diarrhea.

Regarding chemotherapy-induced intestinal mucositis [60,61,62,63,64,65,66,67,68] (Table 3), all the studies we have found used the same oligomeric formula of enteral nutrition (Elental^®^). In total there are 345 patients, of whom 203 were treated with this oligomeric enteral nutrition, and the rest are controls without enteral nutrition. In summary, the main objective of the studies is the effect of this type of oral nutritional supplement on oral mucositis in patients with chemotherapy. In all of the studies [60,61,62,63,64,65,66,67,68] oral mucositis improves, while the rest of the outcomes show different results. Only one study observes improvement in controls on body weight [68], while two others do not find differences [64,67], although in another they find improvement in lean body mass [60]. With regards to the response of diarrhea to this supplementation, the majority of studies do not find differences compared to the control group [62,65,66,68]. Despite these poor results, some authors note that hospitalization time decreases [62], and there are fewer cases in which chemotherapy should be stopped due to its side effects [64,67].

The results of the studies that we have reviewed are very heterogeneous, both in the type of patients, as well as the type of cancer to be treated, in the chemotherapy/radiotherapy dose and in the enteral nutrition formula used, thus the difficulty to draw conclusions. It is not known what proportion of normal diet should be replaced by oligomeric formula to confer the most benefit. In addition to the proportion determination, it is likely that the formulation of the oligomeric diet is important. Similarly, the proportion of fat derived from medium chain triglycerides is likely to be impactful, as this will influence the degree of pancreatic and biliary secretions, both of which may exacerbate damage to the intestinal mucosa [45].

Although oligomeric enteral nutrition formulas have not shown homogeneous efficacy in all studies, we do not have effective pharmacological treatments for these patients [69,70]. At present, the recombinant keratinocyte growth factor-1, palifermin, is thought to be a promising agent for the management of oral mucositis associated with cancer treatment. Regardless of its usefulness in decreasing oral mucositis, the safety of this growth factor in cancer patients remains unclear, and it is expensive compared to other types of therapies for oral mucositis [71]. Compared to other treatment options available for intestinal mucositis in cancer patients, oligomeric enteral nutrition could be an attractive agent, because it is neither costly, nor a growth factor treatment, such as palifermin. Furthermore, no side effects of enteral nutrition have been reported thus far. However, only a few published reports are available on the efficacy of oligomeric enteral nutrition for the treatment of chemo-radiotherapy-induced intestinal mucositis (Table 2 and Table 3).

Harada et al. [67] show that administration of an oligomeric enteral formula was associated with a suppressed expression of protein C reactive. According to in vitro data from the same group, this oligomeric enteral formula could successfully downregulate the expression of inflammatory cytokines in the immortalized human keratinocyte cell line HaCaT [72]. In this way, it might suppress protein C reactive expression via the downregulation of inflammatory cytokines. Harada et al. also showed previously that this oligomeric enteral formula can treat mucositis and dermatitis by accelerating mucosal and skin recovery through FGF2 induction and reepithelization in vivo [73].

The Tanaka group have studied and measured the intestinal mucosal integrity on the basis of plasma diamine oxidase activity. In a recent study they found that the integrity of the intestinal mucosa tends to be maintained in the enteral oligomeric nutrition group [66]. Previous studies have shown that amino acids themselves can protect the mucosa and have anti-inflammatory effects [74]. Furthermore, the administration of an oligomeric enteral nutrition during chemotherapy has been reported to have the potential prevent mucositis [61,63]. In these studies authors group observe that body weight is maintained in the intervention group, in front of weight loss in a control group.

It could be explained, because oligomeric enteral nutrition with Glutamine may have maintained gastrointestinal mucosal integrity, and in this way, maintain nutrient absorption. Furthermore, amino acids may be more suitable than proteins to provide nutrients, as proteins require digestion. Fijlstra et al. [75] observe that enteral administration of amino acid mixtures enable normal amino acid absorption in patients with mucositis.

Enteral oligomeric nutrition could offer mechanisms of protection of the intestinal mucosa. It might increase and stabilize the intestinal bacterial microbiota [76]. In addition, enteral oligomeric nutrition contains amino acids instead of whole protein. These amino acids could have less antigenic power, and thus have less pro-inflammatory effect for the intestinal mucosa [48]. Therefore, this type of oligomeric enteral nutrition, in addition to providing the necessary nutrients in a malnourished patient, could have a non-negligible, anti-inflammatory effect.

However, in the clinical practice guidelines of the different scientific societies, there are dietary recommendations for the nutritional treatment of OTRD, and in some cases these recommend the use of oligomeric enteral nutrition [22,27], but do not offer protocols of action with oral enteral nutrition.

Based on these nutritional recommendations of the scientific societies and the publications reviewed with oligomeric enteral nutrition formulas, we propose a protocol of action before the appearance of OTRD.

## 4. Nutritional Support Algorithm in Patients with OTRD

A direct effect of OTRD is that malabsorption and therefore the risk of malnutrition in these patients is very high. An oligomeric enteral nutrition diet can be the nutritional treatment of choice in patients with OTRD for its easy absorption. Enteral oligomeric nutrition may also have anti-inflammatory effects, change the microbiota, and also lower antigenic effects that protect the oro-intestinal mucosa [67,68]. Its main effect, however, is to maintain the nutritional status of the patient, and to better withstand oncological treatment.

The algorithm of action that we propose covers two aspects: On the one hand, the nutritional assessment of the patient with regard to its nutrient reserve, and on the other hand, the intestinal capacity to absorb nutrients (Figure 2).

The first step in the algorithm we propose is the nutritional assessment of the OTRD oncologic patient. These patients display a high risk of malnutrition, suffering from cancer cachexia, and under aggressive oncological treatment they present OTRD, a side effect causing malabsorption [10]. There are multiple nutritional assessment tests specific to the patient with cancer [15], the majority resulting in a handful of indicators such as: Non-volitional weight loss, low body mass index or reduced muscle mass, reduced food intake or assimilation, and inflammation or disease burden. These criteria have been gathered in a recent consensus by the Global Leadership Initiative on Malnutrition (GLIM), composed by representatives of several of the leading global clinical nutrition societies, and are those that we recommend [77].

The second step is the assessment of the functional capacity of the patient’s intestine. In this step, the presence of OTRD would indicate the intestinal inability to absorb nutrients and the loss of fluids and electrolytes.

With these two aspects evaluated, we could encounter a case within the most advantageous diagnosis of a well-nourished patient without diarrhea, in which only regular dietary measures to prevent the risk of malnutrition would be necessary. On the opposite side of the spectrum, we would find a patient with severe malnutrition and diarrhea, where a nutritional treatment using exclusive enteral nutrition with an oligomeric formula would be required. Thus, the therapeutic possibilities will vary from less to more complexes, depending upon the patient’s initial nutritional situation and the existence of OTRD. These courses of treatment will range from the usual dietary recommendations, to supplementation with oral oligomeric enteral nutrition, complete enteral nutrition with oligomeric formula and then to finally total parenteral nutrition.

After an initial evaluation and nutritional therapy, a reassessment after 5–7 days of the nutritional treatment effectiveness, both OTRD responses and nutritional parameters, is recommended. Consequently, if symptoms do not subside, or the patient continues to deteriorate, it is recommended to move to the next therapeutic step.

## 5. Conclusions

Oncology therapies cause frequently digestive toxicity, such as nausea, vomiting, mucositis and diarrhea. OTRD and malnutrition tend to manifest together in the oncological patient as a consequence of the disease itself, or because of the treatments that are administered. Regardless of the cause of the OTRD, it is essential to carry out a nutritional treatment that facilitates absorption, reduces waste and digestive symptoms, and contributes to the prevention of malnutrition, which is so frequent in the oncological population. Formulas containing peptides and medium chain triglycerides can facilitate absorption in case of, e.g., malabsorption or short bowel syndrome. Standard OTRD management guidelines are widely published. Most of the treatments that appear in the guidelines are focused upon the treatment of OTRD, but very few consider the adequate nutritional support in order to prevent malnutrition. Some guidelines recommend the use of oligomeric enteral nutrition, but do not offer any protocols of action with oral enteral nutrition. The results of the studies that we have reviewed are very heterogeneous, so it is very difficult to draw conclusions. Nevertheless, we consider that the use of nutritional supplements with oligomeric formulas in patients with OTRD who present malnutrition could be very useful. Because enteral oligomeric nutrition could offer protective mechanisms for the intestinal mucosa, we propose a protocol of action before the appearance of OTRD.

Although this review is based mainly upon studies focused on chemotherapy and radiotherapy, we think that our conclusions and recommendations would most likely be widely applied to the management of malnutrition associated to new anticancer agents, despite the paucity of data about their management.

## Figures and Tables

**Figure 1 nutrients-11-01888-f001:**
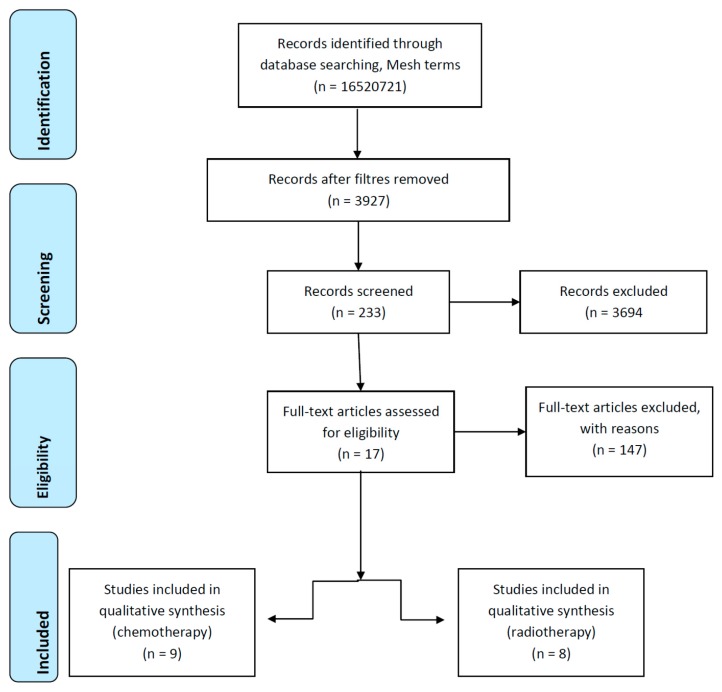
Flow chart describing the literature search process.

**Figure 2 nutrients-11-01888-f002:**
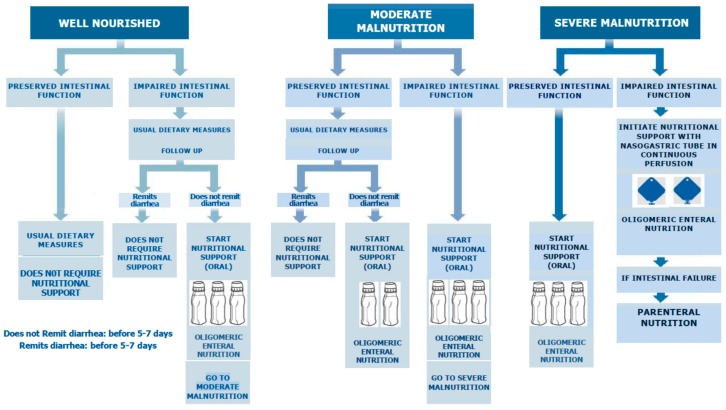
Nutritional support algorithm in patients with Oncology treatment-related diarrhea.

**Table 1 nutrients-11-01888-t001:** Clinical Practice Guidelines for the management of oncology treatment-related diarrhea (OTRD).

Author, Year	Oral HidratatIon	LowF	Low Fiber	Soluble Fiber	Small, Frequent Meals	Avoid Spicy, Fried Foods	Avoid Insoluble Fiber	Avoid Very Hot or Cold Foods/Fluids	Avoid Sorbitol-coNtaining Substances	Avoid Alcohol, Caffeine	Limiting Lactose-coNtaining Products	Enteralor ParenteraLnutrition
**Saltz, LB (2003)** [19]	3–4 L water, clear soup, noncarbonated soft drinks	x	x			x	x				x	Supplementation with the amino acid glutamine
**Benson, Al B. (artículo especial) (2004)** [20]	8–10 glasses of clear liquids at day		x		x					x	x	Avoid high-osmolar supplements
**Canadian Working Group (2007)** [21]	Hydration oral	x								x	x	
**Lupiañez Y. (2008).** [22]	Fluids noncarbonated, nor irritants	x	x	x						x	x	Indicated in diarrhea (oligomeric formula)
**BC Cancer Agency (2010)** [23]	10–12 cups of fluids throughout the day	x	x	x	x	x	x	x	x	x	x	May be indicated in GRADE 3 or 4 diarrhea
**Stein, A (2010)** [24]	Drink 8–10 large glasses of clear liquids a day (eg, Gatorade or broth)				x					x	x	
**Pan Birmingham Cancer Network (2012)** [25]	Drink plenty of fluids.		x			x	x			x	Temporarily to see if this improves symptoms	
**Shaw, C (2012)** [26]	three to four liters of fluid per day	x			x	x				x		
**Wedlake, L.J. (2013)** [27]		should not be recommended	should not be recommended								Should not be recommended	Total replacement of diet with elemental formula may be appropriate in severe toxicity.
**McQuade, RM (2014)** [28]	Drink 8–10 large glasses of clear liquids a day				x						x	
**Andreyev, J (2014)** [29]	Five sachets in 1 L water (consider 8–10 sachets in 1 L is for replacing electrolyte deficits)									x	Especially if the diarrhea is accompanied by marked bloating	
**ESMO Clinical Practice Guidelines (2015)** [30]						x		x		x		Probiotics to prevent diarrhea in a pelvic malignancy
**North of Scotland Cancer Network (2015)** [31]	Increase oral fluids (2–3 L per day),	x	x				x			x	x	
**Oxford University hospital NHS foundation trust (2015)** [32]	Fluids appropriate for weight/age	x										
**Grabenbauer, G (2016)** [33]						x		x		x		Enteral nutrition with severe malnutrition or no enteral food intake for >7 days or insufficient intake
**Mardas, M. (2017)** [34]	Fluids isotonic	x	x	x		x			x	x	x	Avoid high osmolar dietary supplements
**Ecancer medical science (2017)** [35]	Fluids											
**Thompson, KL (2017)** [36]	Hydration oral											Dietary supplements containing EPA or MFS (fish oil), glutamic, vitamins
**Karin, J. (2017)** [37]	High fluid intake			x						x	x	
**PDQ Supportive and Palliative Care. NCI (2018)** [38]	Liquid intake to at least 3 L per day (e.g., water, sports drinks..)avoid beverages.	x	x	x	x	x	x	x	x	x	x	
**Thomsen, M (2018)** [39]	Fluids											Activated charcoal, glutamine, and probiotics
**Oncolink (2018)** [40]	8–10 glasses of clear liquids per day				x				x	x	x	
**ESMO Clinical Practice Guidelines (2018)** [41]	Diluted fruit juices and flavored soft drinks along with saltine crackers and broths or soups					x	x			x	There is insufficient evidence	Glutamine and omega fatty acids. Requires more studies.

Low fiber diet (e.g., white rice and bread, applesauce); Soluble fiber (e.g., fruits and vegetables without skins, oat bran, barley); Insoluble fiber (e.g., skins of fruits and vegetables, wholegrain and multigrain foods).

**Table 2 nutrients-11-01888-t002:** Studies with oligomeric diet and Radiotherapy-induced oro-intestinal mucositis.

Author, Year	Type of Study	Number Patients, Sex, Age (y)	Disease	Intervention	Follow-Up	Component Diet	Object of Study	Results	Confounders
Brown [52] 1980	One single-centre parallel group unblinded RCT	68 patients (16 in control group, 30 who adhered to the ON diet and 21 who did not adhere to the full period of supplementation). Men 70%, 69–29y.	Mainly urological tumours, with many other types of primary tumours.	Compare reduced fibre diet (control) versus three sachets of ON with a reduced fibre diet.	During irradiation	Vivonex HN (Eaton Lab): 900 kcal and 40 g of mixed amino acids, simple sugars, fats, vitamins and mineral.	Reported weight and stool frequency.	Mean stool frequency 4/d for all three groups. Mean weight loss was 1.6 kg, 0.5 kg and 1.7 kg respectively.	Heterogeneous group. Only means were reported with no measurement of variance given
Foster [53] 1980	One single-centre parallel group unblinded RCT	32 patients (12 control and 20 intervention). 75% male, 66y,	Carcinoma bladder, prostate, uterus and testis	Compare reduced fibre diet (control) versus three sachets of Vivonex HN with a reduced fibre diet.	During irradiation	Vivonex HN (Eaton Lab): 900 kcal and 40 g of mixed amino acids, simple sugars, fats, vitamins and mineral.	Investigate the metabolic effects of therapeutic irradiation in different diet planning	Weight loss similar (1.4 vs. 1 Kg). No differences in metabolic and hormonal determinations.	Few of the patients were underweight at the start of treatment.
Beer [54] 1985	One single-centre, prospective cohort study	8 patients (7 female, 1 male). 3 control and 5 intervention.	Gynecologic or testicular malignancy	Compare oral diet versus vivonex-HN and Criticare-HN	Five days	Vivonex HN (Eaton Lab): 900 kcal and 40 g of mixed amino acids, simple sugars, fats, vitamins and mineral. Criticare HN: 900 kcal and 38 g of protein hydrolyzed casein, amino acids	Investigate weight and stool frequency	The mean daily fecal weight and energy were decreased	Few patients, The follow-up is very short
McArle [55] 1986	One single-centre parallel group unblinded RCT	56 patients (32 control and 24 intervention). 75% men, 65y	Invasive bladder cancer	Control: regular hospital diet or TPN during RT. Intervention: 1600–2000 Kcal during 3 days before RT and during RT supplementation with ON continuous feeding nasoduodenal tube or by mouth.	During irradiation	Vital HNR (Ross Lab): Partially hy drolyzed whey, meat, soy, hydrolyzed cornstarch and sucrose, safflower oil and MCT.	Investigate whether elemental diet could prophylaxis intestinal injury by radiotherapy.	The severity of the diarrhea in control was significantly greater. No bloody diarrhea. Positive nitrogen balance. Biopsy ileal normal mucosa morphologia and maintenance enzyme activity.	Control group is retrospective dates.
Craighead [56] 1998	One single-centre uncontrolled phase II prospective cohort study	61 patients (16 intervention group and a cohort of 45 patients (control)	primary cervical and endometrial cancer patients	Intervention with ON. Both cohorts diet restricted lactose, low fibre (12g daily), moderate fat intake (< 30% calories), adequate protein and carbohydrate. Intake of fruits, caffeine and other bowel stimulants restricted.	From three days before the start of radiotherapy to the last day of radiotherapy	Vital HNR: Partially hy drolyzed whey, meat, soy, hydrolyzed cornstarch and sucrose, safflower oil and MCT.	Assess compliance elemental supplements and preliminary assess of the efficacy.	Intervention vs. control: RTOG grade (55% vs. 15%, p < 0.001); Mean (SD) duration of diarrhoea during treatment was 5.85 days (4.44) vs. 12.2 days (6.95).	The study was small and not powered for GI symptoms or weight. Neither randomised nor controlled. Diarrhoea was not clearly defined. It was not clear whether there were any differences between baseline characteristics of the cohorts.
Capirci [57] 2000	Conference abstract of a multicentre RCT	677 patients (332 intervention group and 345 control group).	439 with primary rectal cancer, 228 primary uterine cancer and 10 prostate cancer.	Natural diet plus elemental diet compared with standard diet (control)	Not defined	OD Not defined.	The outcomes were RTOG score and change in weight pre- and post-radiotherapy.	Toxicity significantly less in intervention group. 12 vs. 44 patients required a break in radiotherapy due to GI toxicity in the intervention and control group. The total numbers with each grade of diarrhoea were not reported. Weight response significant differences intervention vs. control: grade 1 toxicity (+1 vs. 0 Kg ), grade 2 (0 vs. −1.3 Kg) and grade 3–4 (−5.5 vs. −4 Kg)	There is no full report. Available data limited. It was not clear whether change in weight reported was mean change in weight and no measure of variance was reported. The statistical methods not described. Heterogeneous group.
McGough [58] 2008	One single-centre parallel group unblinded RCT	50 patients (29 women and 21 men). Intervention (25 people), Control (25 people)	Gynaecological, urological or lower GI malignancy	Control: habitual diet. Intervention: Replace one meal per day (33% of daily calories) with ON.	The first three weeks of radiotherapy.	EO28 Extra (SHS Lab): 85 Kcal, 14% protein, 35% fats (MCT 35%), 52% carbohydrate.	Primary outcome was GI symptoms at week five using IBDQB index. Other outcomes: GI symptoms using VIQ and RTOG toxicity grade, Weight and BMI, and Faecal Calprotectin	GI symptoms increased between baseline and both weeks three and five (P value < 0.001) in both groups. VIQ scores improved for patients in the intervention group comparing week 10 versus week five (P value < 0.001) but not in the control group (P value = 0.06). there was poor compliance with the intervention (6/25, 24%) and no improvement was seen in terms of GI symptoms or nutritional status.	Different baseline characteristics in terms of primary tumour and treatment regimen, The median intake of elemental diet, was lower than the prescribed volume required to provide 33% of caloric requirement. Six patients were non-compliant
Feng Shao [59] 2013	One single-centre parallel group unblinded RCT	46 patients (24 treatment group and 22 control group,). (22 males and 24 females), 60.2y.	Abdominal tumor with post-radiation enteritis	Intervention: Peptisorb plus triple live microorganismal tablets, L-Glutamine enteric capsule and fish oil capsule. Control: Peptisorb only	Whenrceiving radiotherapy or within 3 weeks post-radiotherapy	Peptisorb: ON 16 % protein (85 % small peptides and 15 % amino acid peptide), 9 % fat and 75 % carbohydrates. Microorganismal agent comprised Bifidobacterium, lactobacillus and Streptococcus thermophilus.	Investigate the effect of microbial immune enteral nutrition by microecopharmaceutics and deep sea fish oil and glutamine and Peptisorb on the patients with acute radiation enteritis in bowel function and immune status.	Intervention group: Abdominal pain, bloating and diarrhea was better than the control group (P values were 0.018, 0.04 and 0.008 after 7 days; P values were 0.018, 0.015 and 0.002 after 14 days); and the cellular immune parameters were better than the control group(P = 0.008, P = 0.039, P = 0.032);	Heterogeneous group. Difficult know if the results are because, mirocoorganism, glutamin o or fish oil. Both use ON.

ON: Oligomeric nutrition; GI: Gastrointestinal; RTOG: Radiation Therapy Oncology Group criteria; VIQ: Vaizey Incontinence Questionnaire; TPN: Total parenteral nutrition.

**Table 3 nutrients-11-01888-t003:** Studies with oligomeric diet and chemotherapy-induced oro-intestinal mucositis.

Author, Year	Type of Study	Numbrer patientes, Sex, Age (y).	Disease	Intervention	Follow-Up	Component Diet	Object of Study	Results	Confounders
Ishikawa [60] 2016	Randomized, open label, phase 2 clinical trial	33 patients (17 azulene and 16 OD)	Primarysquamous cell carcinoma of the esophagus who were scheduled to undergo chemotherapy or chemoradiotherapy	Randomized to receive either azulene oral rinse (Arm 1) or OD (Arm 2)	During the treatment cycle (4 weeks).	Elental^®^ (Ajinomoto Pharmaceutical) (80 g/ 300 kcal amino-acid-rich, fat free, elemental diet)	Examine the preventive effects of OD on oral mucositis and sarcopenia progression during chemo (chemoradio) therapy for esophageal cancer	OD tended to reduce the incidence of oral mucositis (Arm 1, 23.5% and Arm 2, 12.5%), but there was no statistically significant difference. The average body mass index and body fat mass decreased significantly in both groups. Lean body mass was reduced in Arm 1, but was increased in Arm; the relative change of lean body mass after the treatment was significant between Arm 1 and Arm 2 (p = 0.007). The incidence of diarrhea was greater in Arm 2 than in Arm 1 (31.3 and 11.8%, respectively), and was grade 2 or less.	Single institution, small patient number
Tanaka [61] 2016	One single-centre parallel group unblinded RCT	30 patients (10 control, 10 Glutamin group and 10 Glutamin plus OD). 90% female. Age 68–75y.	Esophageal cancer	Control (no treatment), Gln group (oral 8910 mg Gln/day), Gln plus OD group.	Oral administration of Glutamin and OD began 1 week before chemotherapy and continued during treatment.	Elental^®^ (Ajinomoto Pharmaceutical) (80 g/ 300 kcal amino-acid-rich, fat free, elemental diet)	Investigate the effect of glutamine and OD chemotherapy-inducOD oral mucositis in esophageal cancer patients	The incidence of grade ≥2 oral mucositiswas 60% in control group, 70 % in Glutamin group, and 10 % in Glutamin plus OD group. The percentage of change in body weight and diamine oxidase activity from before chemotherapy was higher in Glutamin plus OD group than control group.	Small number of patients. Effects not well explained.
Morishita [62] 2016	One single-centre prospective cohort study	73 patients (23 autologous HSCT and 50 allogeneic). 21 Control and 52 intervention. Age 47y (17–67). Male 50% intervention and 70% control.	hematopoietic stem cell transplantation (HSCT)	The first 21 patients did not receive OD and in the successive 52 patients received OD	Oral OD was started before conditioning and was continued until 28 days after transplantation	Elental^®^ (Ajinomoto Pharmaceutical) (80 g/ 300 kcal amino-acid-rich, fat free, elemental diet)	The primary endpoint: hospitalization period. The secondary endpoint: occurrence of oral mucositis, nausea, diarrhea and fever.	The median hospitalization period was significantly shorter in intervention group compared to control (34 days vs. 50 days; p = 0.007). Grade 3–4 oral mucositis occurred less in intervention than control (25% vs. 48%; p = 0.06). There were no significant differences in the use of opioid agents, grade 3–4 diarrhea, or grade 3–4 nausea, for both frequency and duration	Single institution, small patient number, prospective cohort study and included both autologous and allogeneic patients.
Ogata [63] 2016	One single-centre prospective pilot study	22 patients (10 male and 12 Female) 67y	Metastatic colorectal cancer patients after developing grade 1–3 oral mucositis.	The OD Elental^®^ (80 g/300 kcal or more per day) was given perorally in addition to normal oral ingestion, together with chemotherapy in each course lasting 2 to 3 weeks (on days 1–14 or days 1–21).	Introduction of OD after developing grade 1–3 oralmucositis	Elental^®^ (Ajinomoto Pharmaceutical) (80 g/ 300 kcal amino-acid-rich, fat free, OD)	Evaluate the preventive effects of Elental^®^, on chemotherapy-induced oral mucositis in colorectal cáncer.	The maximum grade of oral mucositis decreased in 18 of the 22 patients during the first treatment course with Elental^®^ (p = 0.0002) and in 20 of the 22 patients in the second course (p < 0.0001).	Single institution, small simple size, no control group.
Harada [64] 2016	One single-centre retrospective study	74 patients (37 intervention and 37 control group)	Oral squamous cell carcinoma (OSCC) patients who underwent radiation with/without chemotherapy	Recorded data on a daily basis and compared the data from 37 patients who received OD (intervention group) with those from 37 patients who did not receive OD (the control group)	During the treatment period (6 weeks).	Elental^®^ (Ajinomoto Pharmaceutical) (80 g/300 kcal amino-acid-rich, fat free, elemental diet)	Evaluate the preventive effects of Elental^®^, on radiotherapy- or chemoradiotherapy-induced mucositis in OSCC patients.	Most of the patients who consumed OD suffered from a lower degree of mucositis compared to control group. OD was associated with a significantly improved rate of completion of chemoradiation (no interruption). There was no significant difference between OD group and control group in terms of mean change of body weight or total protein and albumin levels in blood serum before and after chemoradiation.	Single institution. Retrospective
Okada [65] 2017	RCT	22 patients (11 intervention and 11 control)	Esophageal cancer undergoing chemotherapy	Intervention: Elental (one pack per day) for 14 days and control not receiving Elental during chemotherapy	During chemotherapy	Elental^®^ (Ajinomoto Pharmaceutical) (80 g/300 kcal amino-acid-rich, fat free, elemental diet)	Evaluate the influence of elemental diet on chemotherapy-induced oral mucositis and diarrhea.	The distribution of the maximum severity of oral mucositis showed a statistically significant reduction in the Elental group (p = 0.020). Regarding diarrhea, no difference was observed between the two groups.	Single institution, small sample size, most patients in the Elental group did not receive the same dose of Elental.
Tanaka [66] 2018	Prospective multi-center feasibility study	19 patients (males 100%), 68y (37–75y).	Stage II/III esophageal squamous cell carcinoma or adenocarcinoma	Orally administerOD OD 2 packs (160 g/day) during chemotherapy.	2 cycles of chemotherapy	Elental^®^ (Ajinomoto Pharmaceutical) (80 g/300 kcal amino-acid-rich, fat free, elemental diet)	The primary was the compliance of an orally administerOD OD 2 packs. The secondary endpoints were the incidence of oral mucositis; the rate of weight fluctuation; plasma diamine oxidase activity; the turnover rate of plasma proteins and adverse events.	70% patients were able to complete the orally administered OD (160 g/day). The incidence of grade ≥ 2 oral mucositis in the OD completion group (15.4%:, 2 of 13 patients) was significantly lower than that in the non-completion group (66.7%, 4 of 6 patients) (p = 0.046). The grade 3 adverse events were: fatigue (15%), fever (15%), anorexia (15%); diarrhea (10%).	Small sample size, No control group,
Harada [67] 2019	One single-centre prospective study	50 patients (25 with OD and 25 control without). Males 70%.	Oral squamous cell carcinoma patients, who received radiation (60-70 Gy) with/without chemotherapy.	Intervention: 1 bottle/day of OD (80 g, 300 kcal) orally. Control without OD.	The median follow up period was 23 (8–37) months	Elental^®^ (Ajinomoto Pharmaceutical) (80 g/300 kcal amino-acid-rich, fat free, elemental diet)	Evaluate the preventive effects of OD on radiotherapy- or chemoradiotherapy–induced mucositis in oral squamous cell carcinoma patients	Multivariate analysis indicated that most of the patients who received Elental^®^ suffered from a lower degree of mucositis and showed significantly improved rate of completion of chemoradiation (no interruption) compared to the control group. There was a significant difference between the Elental^®^ group and the control group in mean change of C-reactive protein levels in blood serum; however, there was no significant difference in mean change of body weight and total protein level in blood serum before and after chemoradiation. Diarrhea were no determinated.	Single institution,
Toyomasu [68] 2019	One single-centre randomizOD open-label study	22 patients (11 control and 11 intervention). Ages ranged from 59 to 80y.	Patients who underwent adjuvant chemotherapy for gastric cancer	One pack of OD per day	During adjuvant chemotherapy	Elental^®^ (Ajinomoto Pharmaceutical) (80 g/300 kcal amino-acid-rich, fat free, elemental diet)	Whether an oral OD prevents chemotherapy associated oral mucositis and body weight loss.	The incidence of oral mucositis was significantly lower in the treatment group (9.1 %) than in the control group (27.3%). The median body weight loss in the treatment group was significantly smaller than that in the control group (P = 0.015). The treatment group was significantly associated with high cumulative S-1 continuation rates (log-rank p = 0.047). No differences in diarrhea between groups.	Single-institutional study with a small sample size

OD: Oligomeric diet.

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
