# Peer review of "Oligomeric Enteral Nutrition in Undernutrition, due to Oncology Treatment-Related Diarrhea. Systematic Review and Proposal of An Algorithm of Action"

_nutrients, 2019, doi:10.3390/nu11081888_

Round 1
Reviewer 1 Report
This article reviews oligomeric enteral nutrition in patients with undernutrition due to systemic treatment induced diarrhea. It is straightforward, well written, and concise and proposes a relevant algorithm of action. Definitely deserves to be published and is a valuable contribution to the “Nutrients” Journal.
Some minor flaws could be addressed before publication.
Minor points:
[1] Lines 92-93: In the section of diarrhea induced by targeted agents, please make a comment about the TKI imatinib, the proteasome inhibitor bortezomib, and the mTOR inhibitor temsirolimus [Boussios S, et al. Systemic treatment-induced gastrointestinal toxicity: incidence, clinical presentation and management. Ann Gastroenterol. 2012;25(2):106-118].
[2] Lines 166-167: Nutrition support can be given to patients through enteral nutrition or parenteral nutrition. Please, make a comment about the evidence that parenteral nutrition is not related to prolong survival, increased nutrition support complications, or major complications as compared to enteral nutrition in cancer patients [Chow R, et al. Enteral and parenteral nutrition in cancer patients: a systematic review and meta-analysis. Ann Palliat Med. 2016;5(1):30-41].
[3] Lines 225-228: A comment about docetaxel-induced enterocolitis, characterized by abdominal pain, diarrhoeas and mucositis should be made [Boussios S, et al. Docetaxel-induced enterocolitis: a serious and potentially fatal adverse event. J BUON. 2011;16(4):778-9].
Author Response
Minor points:
[1] Lines 92-93: In the section of diarrhea induced by targeted agents, please make a comment about the TKI imatinib, the proteasome inhibitor bortezomib, and the mTOR inhibitor temsirolimus [Boussios S, et al. Systemic treatment-induced gastrointestinal toxicity: incidence, clinical presentation and management. Ann Gastroenterol. 2012;25(2):106-118].
Thank you for your bibliographic contribution. Imatinib is already included in the references (13) to the prevalence of diarrhea of the new antioncological drugs in line 94.
Boussios S, Pentheroudakis G, Katsanos K, Pavlidis N. Systemic treatment-induced gastrointestinal toxicity: incidence, clinical presentation and management. Ann Gastroenterol. 2012;25(2):106-118
[2] Lines 166-167: Nutrition support can be given to patients through enteral nutrition or parenteral nutrition. Please, make a comment about the evidence that parenteral nutrition is not related to prolong survival, increased nutrition support complications, or major complications as compared to enteral nutrition in cancer patients [Chow R, et al. Enteral and parenteral nutrition in cancer patients: a systematic review and meta-analysis. Ann Palliat Med. 2016;5(1):30-41].
Thank you for your input. It is a very important point that must always be kept in mind in the nutritional treatment. We have included this clarification (17) on line 109.
“In patients with cancer, parenteral nutrition has not been shown to prolong the survival of patients and also increases the prevalence of complications compared to enteral nutrition”.
Chow R, Bruera E, Chiu L, Chow S, Chiu N, Lam H, McDonald R, DeAngelis C, Vuong S, Ganesh V, Chow E. Enteral and parenteral nutrition in cancer patients: a systematic review and meta-analysis. Ann Palliat Med. 2016 Jan;5(1):30-41. doi: 10.3978/j.issn.2224-5820.2016.01.01.
[3] Lines 225-228: A comment about docetaxel-induced enterocolitis, characterized by abdominal pain, diarrhoeas and mucositis should be made [Boussios S, et al. Docetaxel-induced enterocolitis: a serious and potentially fatal adverse event. J BUON. 2011;16(4):778-9].
Thank you very much for your support. We think it is important in our review to remember the antioncological drugs that are associated with digestive complications.
We add your contribution on line 231.
Boussios S, Pentheroudakis G, Kamina S, Katsanos K, Pavlidis N. Docetaxel-induced enterocolitis: a serious and potentially fatal adverse event. J BUON. 2011 Oct-Dec;16(4):778-9.
Reviewer 2 Report
This article is well written
I have several minor points to be addressed
The abstract should be rewritten in a less descritive style and indicate what the authors found, and what they recommend based upon their trawl of the literature The search strategy is not at all clear. Can the authors clearly indicate what Boolean strategy they used ie. what were the keywords, and how combinations were done using "AND","OR". This is not clear currently. This may affect the numbers in the PRISMA diagram Has this been registered as a systematic review?Author Response
I have several minor points to be addressed
The abstract should be rewritten in a less descritive style and indicate what the authors found, and what they recommend based upon their trawl of the literature.
You are right in your comments. With the limitations in the number of words for the abstract, we add the two aspects that we think are most important in the work. We add in the abstract our conclusions of the literature review carried out, as well as a summary of the algorithm we propose.
Oncology treatment related diarrhea and malnutrition appear together in oncological patients because of the disease itself or the treatments that are administered for it. Therefore it is essential to carry out a nutritional treatment. Enteral nutrition formulas, containing peptides and medium chain triglycerides, can facilitate absorption in cases of malabsorption. There are few references to the use of enteral nutrition in the clinical society guidelines of patient management with OTRD. A bibliographic review of the studies with oligomeric enteral nutrition in OTRD found only 9 studies with chemotherapy (all with the same oligomeric formula in which oral mucositis improves, while the rest of the outcomes show different results) and 8 studies with radiotherapy (with different products and very heterogeneous results). We hereby present our action algorithm to supplement the diet of OTRD patients with an oligomeric enteral nutrition formula. The first step is the nutritional assessment, followed by the assessment of the functional capacity of the patient's intestine. With these two aspects evaluated, the therapeutic possibilities available vary in degrees of complexity: these will range from usual dietary recommendations, to supplementation with oral oligomeric enteral nutrition, complete enteral nutrition with oligomeric formula to potentially total parenteral nutrition.
.
The search strategy is not at all clear. Can the authors clearly indicate what Boolean strategy they used ie. what were the keywords, and how combinations were done using "AND","OR". This is not clear currently. This may affect the numbers in the PRISMA diagram Has this been registered as a systematic review?
You are right but we have presented the search strategy as we have seen in other articles in Nutrients. Anyway, we rewrite the search as we used it in our Boolean strategy.
For this review, we researched on-line databases PUBMED, MEDLINE, EMBASE and the Cochrane library up to March 2019 using the following terms: (((chemotherapy) or (radiotherapy)) and ((oral mucositis) or (diarrhea) or (peptide diet) or (elemental diet) or ( oligomeric diet)) to collect potentially relevant articles. Animal, not clinical trials and non-adult studies were excluded. “radiation enteritis” OR “chemotherapy induced bowel damage” OR “radiation induced bowel damage” OR “diarrhoea” OR “bowel symptoms” AND ”nutrition” OR “elemental” OR “oligomeric” OR “peptide” OR “enteral nutrition” to colect potentially relevant articles.
Unfortunately, we did not registered our review, as we did not considered it necessary given the main focus of the article is the algorithm and the review is a support to its introduction.
Reviewer 3 Report
The authors did a great job in reviewing the problem associated with oncology treatment related diarrhea which is mostly neglected during treatment of cancer specially during chemotherapy as well as radiotherapy. One of the major side effects of radiotherapy is diarrhea and finally leading to malnutrition.
The authors did a great job is reviewing the literature and making a algorithm of action.
Author Response
Thank you very much for your comments.
We agree that diarrhea associated with cancer treatment is an important clinical problem.
We have carried out this literature review with the aim of designing a treatment algorithm.